# SR-17018 Stimulates Atypical µ-Opioid Receptor Phosphorylation and Dephosphorylation

**DOI:** 10.3390/molecules26154509

**Published:** 2021-07-27

**Authors:** Sebastian Fritzwanker, Stefan Schulz, Andrea Kliewer

**Affiliations:** Department of Pharmacology and Toxicology, Jena University Hospital, Friedrich Schiller University Jena, Drackendorfer Straße 1, D-07747 Jena, Germany; Sebastian.Fritzwanker@med.uni-jena.de

**Keywords:** µ-opioid receptor, DAMGO, SR-17018, buprenorphine

## Abstract

Opioid-associated overdoses and deaths due to respiratory depression are a major public health problem in the US and other Western countries. In the past decade, much research effort has been directed towards the development of G-protein-biased µ-opioid receptor (MOP) agonists as a possible means to circumvent this problem. The bias hypothesis proposes that G-protein signaling mediates analgesia, whereas ß-arrestin signaling mediates respiratory depression. SR-17018 was initially reported as a highly biased µ-opioid with an extremely wide therapeutic window. It was later shown that SR-17018 can also reverse morphine tolerance and prevent withdrawal via a hitherto unknown mechanism of action. Here, we examined the temporal dynamics of SR-17018-induced MOP phosphorylation and dephosphorylation. Exposure of MOP to saturating concentrations of SR-17018 for extended periods of time stimulated a MOP phosphorylation pattern that was indistinguishable from that induced by the full agonist DAMGO. Unlike DAMGO-induced MOP phosphorylation, which is reversible within minutes after agonist washout, SR-17018-induced MOP phosphorylation persisted for hours under otherwise identical conditions. Such delayed MOP dephosphorylation kinetics were also found for the partial agonist buprenorphine. However, buprenorphine, SR-17018-induced MOP phosphorylation was fully reversible when naloxone was included in the washout solution. SR-17018 exhibits a qualitative and temporal MOP phosphorylation profile that is strikingly different from any other known biased, partial, or full MOP agonist. We conclude that detailed analysis of receptor phosphorylation may provide novel insights into previously unappreciated pharmacological properties of newly synthesized MOP ligands.

## 1. Introduction

Opioids are the most effective drugs for the treatment of severe pain. However, their clinical use in acute and chronic pain is limited by severe adverse side effects such as respiratory depression, constipation, dependence, and development of tolerance [1,2]. Currently, opioid-associated overdoses and deaths due to respiratory depression from prescription opioids are a major public health problem in the US and other Western countries. It is believed that one way to solve this problem may be the development of biased µ-opioid receptor (MOP) agonists. These compounds have been developed based on the hypothesis that selective activation of the G-protein signal pathway via MOP mediates the analgesic effect by avoiding stimulation of ß-arrestin signaling, which is believed to induce adverse opioid effects such as respiratory depression and constipation.

SR-17018 is one of the most recently described G-protein-biased agonists [3,4]. Schmid et al. (2017) demonstrated an extremely high bias factor in different G-protein assays over ß-arrestin 2 recruitment in vitro, and significant separation between antinociception and respiratory side effects in vivo. In addition to the extremely wide therapeutic window, it was reported later that SR-17018 does not produce tolerance in the hot-plate antinociception assay [5]. Furthermore, it was shown that SR-17018 can reverse morphine tolerance and prevent withdrawal via an unknown mechanism of action [5].

In contrast, a more recent study by Gillis et al. (2020) showed that SR-17018 consistently exhibited low intrinsic efficacy across a variety of assays and showed no statistically significant bias towards or away from any G-protein activation. Furthermore, similar kinetics were observed between antinociception and respiratory depressant effects. Similar in vitro results were obtained with the partial agonist buprenorphine, albeit with an increased therapeutic window regarding respiratory depression [4,6,7]. In clinical settings, buprenorphine is used as an alternative to methadone in the treatment of heroin addiction, due to its mixed agonist–antagonist properties [8]. Collectively, these findings suggest that SR-17018 may be similar to buprenorphine and exhibit partial agonistic properties.

In the past decade, we have shown that the phosphorylation barcode of the MOP carboxyl-terminal tail is dependent on agonist efficacy and indicative of ß-arrestin recruitment and receptor internalization [9,10]. High-efficacy MOP agonists like DAMGO and fentanyl induce a robust hierarchical and sequential multisite receptor phosphorylation, whereas low-efficacy agonists like morphine, oxycodone, and buprenorphine, trigger only phosphorylation at Ser^375^. Recent phosphorylation studies with SR-17018 have reported an unusual phosphorylation pattern which is limited to Ser^375^ during the first 20 min of stimulation, corresponding to a low-efficacy agonist [4]. In contrast, incubation for >30 min leads to a multisite receptor phosphorylation, which corresponds to a high-efficacy agonist. Furthermore, SR-17018-induced MOP phosphorylation is driven by GRK2/3 and is naloxone sensitive.

Given its unusual pharmacological profile and unknown mechanism of action, we performed a series of MOP phosphorylation and dephosphorylation experiments in vitro and compared the effects of SR-17018 with the partial agonist buprenorphine.

## 2. Results

SR-17018 was developed as a G-protein-biased MOP agonist, but exhibits a number of pharmacological effects which cannot be explained by the biased signaling hypothesis. To better understand SR-17018 ligand properties, we performed a series of MOP phosphorylation and dephosphorylation experiments comparing SR-17018 to the low-efficacy agonist buprenorphine and the full agonist DAMGO as internal standard.

### 2.1. Agonist-Induced Dose-Dependent MOP Phosphorylation

First, we evaluated dose- and time-dependent MOP phosphorylation induced by DAMGO, SR-17018 or buprenorphine (Figure 1). As shown in Figure 2, 30 min exposure at 37 °C to saturating concentrations of SR-17018 induced a multisite phosphorylation that was indistinguishable from that induced by DAMGO. In contrast, buprenorphine induced only a robust Ser^375^ phosphorylation under otherwise identical conditions.

### 2.2. Agonist-Induced Time-Dependent MOP Phosphorylation

As depicted in Figure 3A, DAMGO-stimulated MOP phosphorylation occurred rapidly within seconds to minutes at RT. S375 is the initial site of a hierarchical phosphorylation cascade. The following phosphorylation at T370, T379, and T376 requires priming S375 phosphorylation [9]. In contrast, both SR-17018- and buprenorphine-mediated MOP phosphorylation required extended exposure times (>20 min). Furthermore, SR-17018 and DAMGO promoted a robust internalization, which resulted in a receptor accumulation in the perinuclear recycling compartment (Figure 3B). In contrast, buprenorphine failed to stimulate any detectable MOP endocytosis (Figure 3B).

### 2.3. PBS Buffer Washout of Agonist-Induced Phosphorylation

Next, we evaluated the temporal dynamics of MOP dephosphorylation after extensive ligand washout with PBS (Figure 4). The DAMGO-induced phosphorylation was quickly reversed within 5 to 10 min after agonist removal. T370 and T379 were dephosphorylated immediately, whereas S375 and T376 dephosphorylation required between 10 and 20 min. In contrast, SR-17018-mediated MOP phosphorylation was retained for hours under otherwise identical conditions (Figure 4B). Similar, the buprenorphine-stimulated Ser^375^ phosphorylation was also resistant to PBS washout (Figure 4A).

### 2.4. Naloxone Washout of Agonist-Induced Phosphorylation

Interestingly, when 10 µM naloxone was included into the washout solution, MOP dephosphorylation was strongly facilitated in SR-17018-treated but not in buprenorphine-treated cultures (Figure 5).

## 3. Discussion

SR-17018 is unique in that it exhibits an atypical MOP phosphorylation and dephosphorylation profile [4]. Saturating concentrations of SR-17018 stimulate a full agonist-like multisite phosphorylation of MOP but with delayed onset (>20 min). Similar slow phosphorylation kinetics are observed with the partial agonist buprenorphine, whereas the full agonist DAMGO induces full MOP phosphorylation within seconds. At least three kinases contribute to agonist-induced MOP phosphorylation namely GRK2, GRK3, and GRK5 [9,11]. The most likely explanation is that SR-17018, buprenorphine, and DAMGO restrain the receptor in different conformations, which exhibit different affinities for individual GRKs [4]. In fact, the selective engagement of different GRKs to differently activated MOP receptors could be a major source of biased signaling as it is the driving force for recruitment of arrestin isoforms 1 and 2 to the receptor [9,10]. Thus, different GRK-mediated phosphorylation patterns should be taken into account in the development of new MOP agonists with beneficial side-effect profiles.

For many years, the biased signaling concept has been reduced to analysis of G-protein signaling versus ß-arrestin 2 recruitment, and the resulting bias factor has been proposed as a predictor of the therapeutic window. SR-17018 is one candidate compound that was developed based on the biased signaling hypothesis [3]. While the initial study reported an extremely high bias factor in different G-protein assays over ß-arrestin 2 recruitment, later work showed no statistically significant bias towards or away from any G-protein activation [4]. Nevertheless, the present study revealed a unique MOP phosphorylation and internalization profile for SR-17018 that does not support the initial report of an extremely high bias factor.

Conversely, dephosphorylation of DAMGO-activated MOPs occurred within minutes after agonist washout. In contrast, SR-17018-stimulated MOP phosphorylation occurred in a delayed manner similar to that observed with buprenorphine and persisted for hours after agonist washout. These data suggest that SR-17018 remains tightly bound to the MOP receptor after washout, while preventing receptor dephosphorylation. However, SR-17018-induced MOP phosphorylation was reversible when the antagonist naloxone was included in aqueous washout solutions. In contrast, buprenorphine-stimulated MOP phosphorylation was not reversible by naloxone. These results predict that SR-17018 has a very slow off-rate at MOP, similar to that known for buprenorphine [4,12]. However, SR-17018 has a much lower affinity than buprenorphine so it can easily be displaced by naloxone [3,12]. 

SR-17018 exhibits a peculiar pharmacological profile in preclinical animal models, where it has been shown to prevent opioid withdrawal signs [3,5]. Such activity has previously been observed for buprenorphine but not for any other biased MOP agonist [8,13]. This suggests that opioids with delayed dephosphorylation kinetics may be useful for opioid maintenance therapy. Nevertheless, SR-17018 differs from buprenorphine in that its effects are easily reversible with naloxone.

## 4. Materials and Methods

### 4.1. Reagents and Antibodies 

[D-Ala^2^, N-Me-Phe^4^, Gly^5^-ol]-Enkephalin acetate salt (DAMGO) was purchased from Sigma Aldrich (Munich, Germany). SR-17018 was obtained from MedChemExpress (Monmouth Junction, NJ, USA), buprenorphine from Indivior (Dublin, Ireland), and naloxone from Ratiopharm (Ulm, Germany). Pierce^TM^ HA epitope tag antibody was obtained from Thermo Scientific (Rockford, IL, USA). The rabbit polyclonal phosphosite-specific µ-opioid receptor antibodies anti-pT370 (7TM0319B), anti-pT376 (7TM0319D), anti-pT379 (7TM0319E), anti-pS375 (7TM0319C) and anti-HA antibody (7TM000HA) were obtained from 7TM Antibodies (Jena, Germany) [9,11,14,15]. The secondary horseradish peroxidase (HRP)-linked anti-rabbit antibody was purchased from Cell Signaling (Frankfurt, Germany).

### 4.2. Cell Culture and Transfection

HEK293 cells were originally obtained from the German Resource Centre for Biological Material (DSMZ, Braunschweig, Germany) and grown in Dulbecco’s modified Eagle’s medium supplemented with 10% fetal calf serum, 2 mM L-glutamine, and 100 U/mL penicillin/streptomycin and cultured in a humidified atmosphere containing 5% CO_2_. Cells were stably transfected with mouse MOP-HA and the assays performed have been extensively characterized in previous publications [9,11].

### 4.3. Western Blot Assay

HEK293 cells stably expressing HA-MOP were seeded onto poly-l-lysine-coated 60 mm dishes and grown to 90% confluency. After agonist stimulation, cells were lysed in RIPA buffer (50 mM Tris-HCl, pH 7.4, 150 mM NaCl, 5 mM EDTA, 1% Nonidet P-40, 0.5% sodium deoxycholate, 0.1% SDS) containing protease and phosphatase inhibitors (Complete mini and PhosSTOP; Roche Diagnostics, Mannheim, Germany). When indicated, cells were washed three times with either PBS buffer (PBS washout) or PBS supplemented with 10 µM naloxone (naloxone washout). After removal of agonist, the cells were incubated in the absence of agonist at 37 °C as indicated and lysed in RIPA buffer containing protease and phosphatase inhibitors, as described previously. The assays were performed at both physiological temperature (37 °C) and at room temperature (22 °C) to slow down the cellular processes if indicated. Pierce^TM^ HA epitope tag antibody beads (Thermo Scientific, Rockford, IL, USA) were used to enrich HA-tagged MOP. The samples were washed several times afterwards. To elute proteins from the beads, the samples were incubated in SDS sample buffer for 25 min at 43 °C. Supernatants were separated from the beads, loaded onto 8% SDS polyacrylamide gels, and then immunoblotted onto nitrocellulose membranes afterwards. After blocking, membranes were incubated with anti-pT370 (7TM0319B), anti-pS375 (7TM0319C), anti-pT376 (7TM0319D), or anti-pT379 (7TM0319E) antibody overnight at 4 °C (7TM Antibodies, Jena, Germany). Membranes were incubated in HRP-linked secondary antibody for 2 h, followed by detection using a chemiluminescence system (90 mM *p*-coumaric-acid, 250 mM luminol, 30% hydrogen peroxide). Blots were subsequently stripped and incubated again with the phosphorylation-independent anti-HA antibody to confirm equal loading of the gels. Protein bands on Western blots were exposed to X-ray films.

### 4.4. Immunocytochemistry

HEK293 cells stably expressing HA-MOP were seeded onto poly-l-lysine coated 24-well plates overnight. On the next day, cells were pre-incubated with anti-HA antibody for 2 h at 4 °C. Cells were then transferred to 37 °C, exposed to 10 µM agonist for 30 min at 37 °C and fixed with 4% paraformaldehyde and 0.2% picric acid in phosphate buffer (pH 6.9) for 30 min at room temperature (RT). After washing the coverslips with PBS *w*/*o* Ca^2+^/Mg^2+^ buffer several times, cells were blocked with phosphate buffer containing 3% NGS for 2 h and were then incubated with Alexa488-conjugated secondary antibody (1:2000) (LifeTechnologies, Thermo Fisher Scientific A11008) overnight at 4 °C. On the next day, cells were washed several times with PBS *w*/*o* Ca^2+^/Mg^2+^ and specimens were mounted with Roti^®^-MountFluorCare DAPI (Carl Roth, HP20.1) and examined using a Zeiss LSM510 META laser scanning confocal microscope (Zeiss, Jena, Germany). 

### 4.5. Data Availability

The authors declare that all data supporting the findings of this study are presented within the paper and its supporting information files. The data that support the findings of this study are available from the authors upon reasonable request. 

## 5. Conclusions

Together, the present study reveals a mechanism of action for SR-17018 that is clearly different from any other known MOP agonist. Our findings also demonstrate that newly synthesized compounds should be fully characterized, including detailed analysis of their receptor phosphorylation kinetics, before classification as biased, partial, or full agonists.

## Figures and Tables

**Figure 1 molecules-26-04509-f001:**
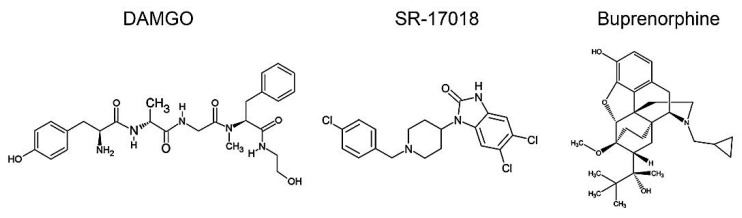
Chemical structures of DAMGO, SR-17018, and buprenorphine.

**Figure 2 molecules-26-04509-f002:**
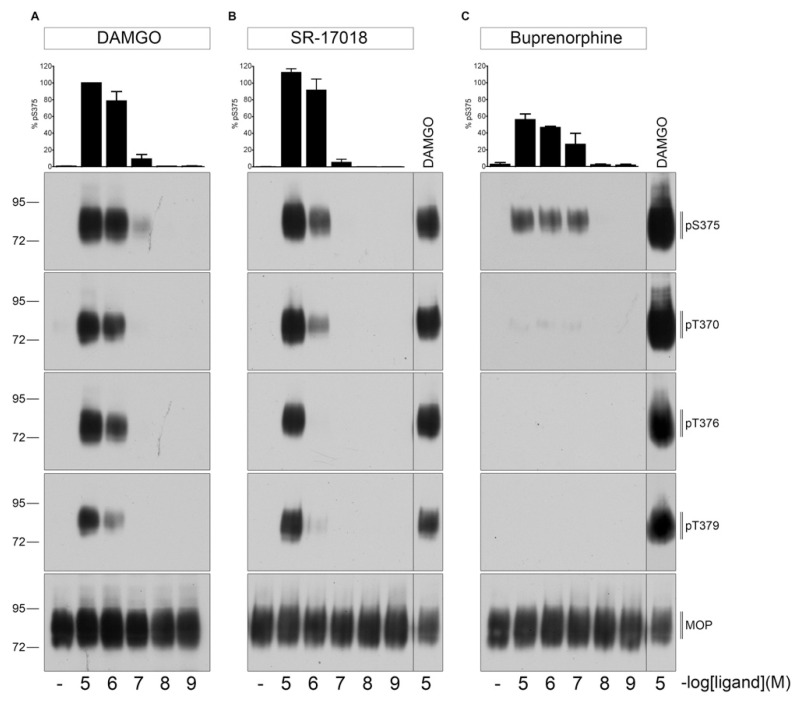
Dose-dependent multisite phosphorylation by DAMGO, SR-17018, and buprenorphine. HEK293 cells stably expressing HA-MOP were either treated with (**A**) DAMGO, (**B**) SR-17018, or (**C**) buprenorphine with concentrations ranging from 10 µM to 1 nM for 30 min at 37 °C. Cells were lysed and immunoblotted with the anti-pT370 (pT370), anti-pT376 (pT376), anti-pT379 (pT379), or anti-pSer375 (pS375) antibodies. Blots were stripped and reprobed with the phosphorylation-independent anti-HA-tag antibody to confirm equal loading of the gels. S375 phosphorylation was quantified (upper panel) and expressed as percentage of maximal phosphorylation in control cells, which was set at 100%. Data correspond to mean ± SEM from three independent experiments. Positions of molecular mass markers are indicated on the left (in kDa). ((**B**,**C**) right lane) 10 µM DAMGO samples were used as a control to visualize different development times on X-ray films. Blots are representative of three independent experiments.

**Figure 3 molecules-26-04509-f003:**
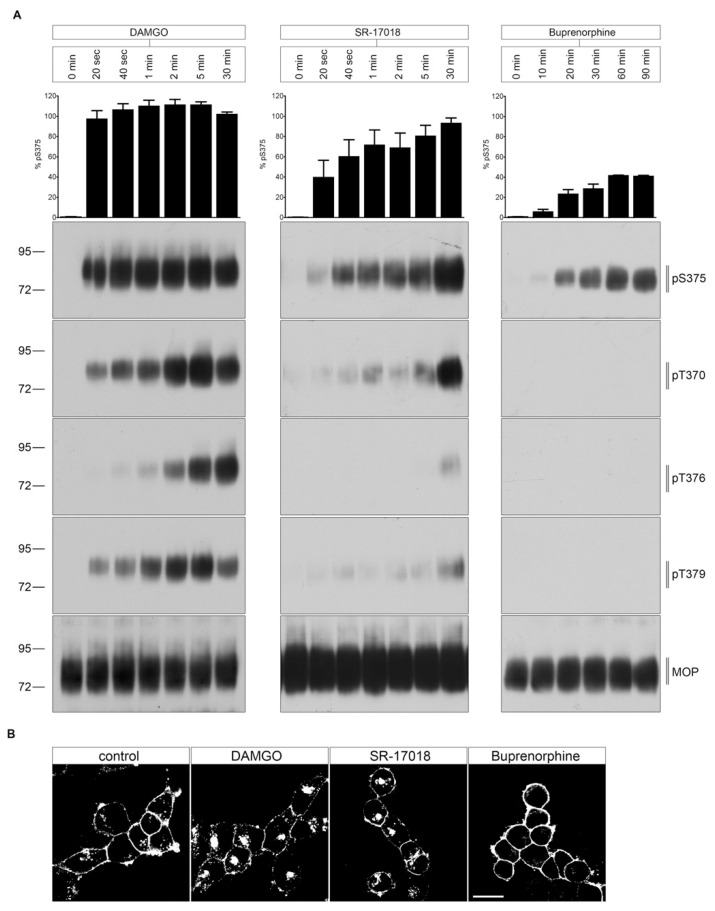
Time course of multisite phosphorylation by DAMGO, SR-17018, and buprenorphine. (**A**) HEK293 cells stably expressing HA-MOP were incubated with (left panel) 10 µM DAMGO, (middle panel) 10 µM SR-17018, or (right panel) buprenorphine for the indicated time periods at RT. Cells were lysed and immunoblotted with the anti-pT370 (pT370), anti-pT376 (pT376), anti-pT379 (pT379), or anti-pSer375 (pS375) antibodies. Blots were stripped and reprobed with the phosphorylation-independent anti-HA-tag antibody to confirm equal loading of the gels (MOP). S375 phosphorylation was quantified (upper panel) and expressed as percentage of maximal phosphorylation in control cells, which was set at 100% (data not shown). Data correspond to mean ± SEM from three independent experiments. Positions of molecular mass markers are indicated on the left (in kDa). Blots are representative of three independent experiments. (**B**) HEK293 HA-MOP cells were pre-incubated with anti-HA antibody for 2 h at 4 °C. Afterwards, cells were treated with either 10 µM DAMGO, 10 µM SR-17018, or 10 µM buprenorphine for 30 min at 37 °C. After fixation, the cells were incubated with Alexa488-conjugated secondary antibody and examined using confocal microscopy. Figure shows representative images of three independent experiments. Scale bar: 20 µm.

**Figure 4 molecules-26-04509-f004:**
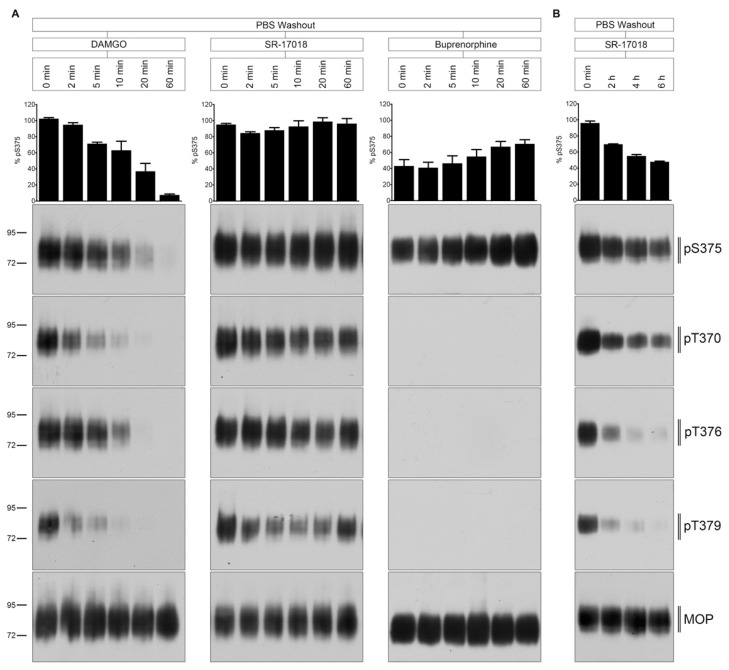
Time course of multisite dephosphorylation after PBS buffer washout. (**A**) HEK293 cells stably expressing HA-MOP were incubated with (left panel) 10 µM DAMGO, (middle panel) 10 µM SR-17018, or (right panel) buprenorphine for 30 min at 37 °C. Cells were washed three times with PBS buffer (PBS washout) and then incubated in the absence of agonist for 0, 2, 5, 10, 20, or 60 min at 37 °C. (**B**) HEK293 cells stably expressing HA-MOP were incubated with 10 µM SR-17018 for 30 min. Cells were washed three times with PBS buffer and then incubated in the absence of agonist for 0, 2, 4, or 6 h at 37 °C. (**A**,**B**) Cells were lysed and immunoblotted with the anti-pT370 (pT370), anti-pT376 (pT376), anti-pT379 (pT379), or anti-pSer375 (pS375) antibodies. Blots were stripped and reprobed with the phosphorylation-independent anti-HA-tag antibody to confirm equal loading of the gels (MOP). S375 phosphorylation was quantified (upper panel) and expressed as percentage of maximal phosphorylation in control cells, which was set at 100% (data not shown). Data correspond to mean ± SEM from three independent experiments. Positions of molecular mass markers are indicated on the left (in kDa). Blots are representative of three independent experiments.

**Figure 5 molecules-26-04509-f005:**
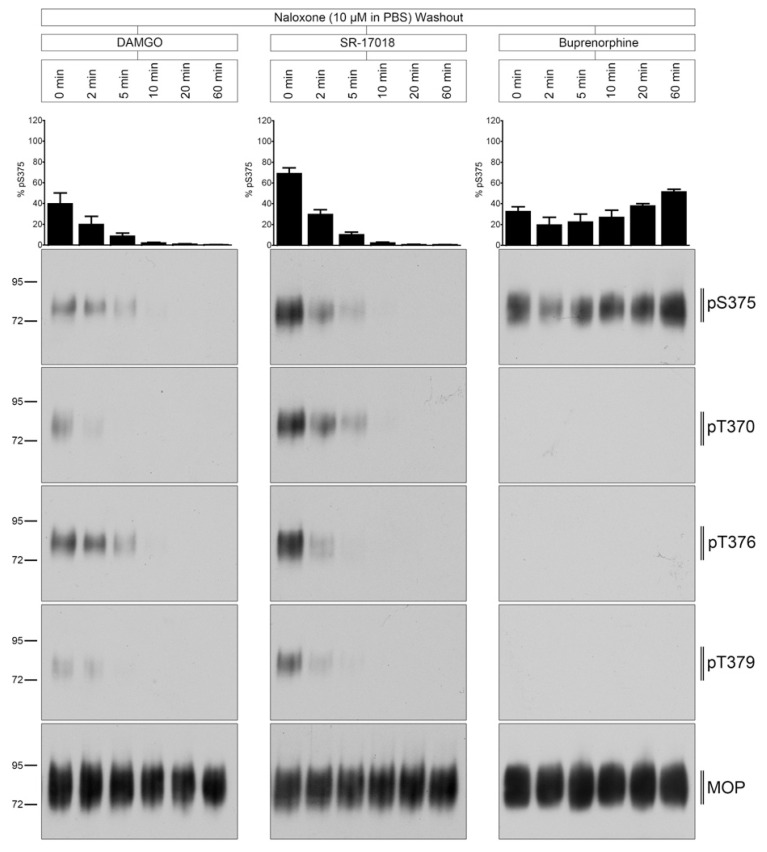
Time course of multisite dephosphorylation after naloxone washout. HEK293 cells stably expressing HA-MOP were incubated with (left panel) 10 µM DAMGO, (middle panel) 10 µM SR-17018, or (right panel) 10 µM buprenorphine for 30 min at 37 °C. Cells were washed three times with 10 µM naloxone and then incubated in the absence of agonist for 0, 2, 5, 10, 20, or 60 min at 37 °C. Cells were lysed and immunoblotted with the anti-pT370 (pT370), anti-pT376 (pT376), anti-pT379 (pT379), or anti-pSer375 (pS375) antibodies. Blots were stripped and reprobed with the phosphorylation-independent anti-HA-tag antibody to confirm equal loading of the gels (MOP). S375 phosphorylation was quantified (upper panel) and expressed as percentage of maximal phosphorylation in control cells, which was set at 100% (data not shown). Data correspond to mean ± SEM from at least three independent experiments. Positions of molecular mass markers are indicated on the left (in kDa). Blots are representative of three independent experiments.

## Data Availability

The data presented in this study are available on request from the corresponding author.

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
