# Peer review of "SR-17018 Stimulates Atypical µ-Opioid Receptor Phosphorylation and Dephosphorylation"

_molecules, 2021, doi:10.3390/molecules26154509_

Round 1

Reviewer 1 Report

SR 17018 is a functionally selective novel, non-peptide opioid ligand for the MOP receptor with efficacious G protein-stimulation (EC50   97 nM) over β-arrestin 2 recruitment (EC50 > 10 μM) ; increases latency to withdraw in the hot plate and warm water tail-flick assay (ED50s  are 6.9 and 7.7 mg/kg, respectively) without inducing respiratory depression in mice at up to 48 mg/kg.  In this compact paper by Fritzwanker et al., examined the phosphorylation pattern of SR 17018 in detail by comparing it with phosphorylation obtained with two well-known opioids, the DAMGO peptide and buprenorphine. Partial versus full agonists and biased agonism which distinguishes two important signaling mechanisms, G-protein activation and beta-arrestin recruitment, are in the focus of receptorology. Based on these, the submitted communication is professionally important and timely. I found the article interesting and high quality so I definitely support its publication.

In order to further increase the value of this communication, I would like to consider, answer and accept the following questions and suggestions.

-- The chemical structure of the title compound should be included.

-- Fig 1. upper panel: the height of the fourth column appears to be high for buprenorphine (right). Compare with the corresponding column height of DAMGO (left). There, the spot of the immunoblotting is darker than that of buprenorphine.

--  In the confocal microscopy images shown in the second figure (Fig. 2), the internalized membrane receptors show a compact precipitation image and appear almost as nuclei. Why is this marking?

-- Fig. 1-4. Spots of one phosphothreonine (pT340) are well visible for buprenorphine (Fig. 1), but are not present in any of the other figures. What could be the reason for this?

Author Response

Reviewer 1 comments

Comments and Suggestions for Authors

SR 17018 is a functionally selective novel, non-peptide opioid ligand for the MOP receptor with efficacious G protein-stimulation (EC50   97 nM) over β-arrestin 2 recruitment (EC50 > 10 μM) ; increases latency to withdraw in the hot plate and warm water tail-flick assay (ED50s  are 6.9 and 7.7 mg/kg, respectively) without inducing respiratory depression in mice at up to 48 mg/kg.  In this compact paper by Fritzwanker et al., examined the phosphorylation pattern of SR 17018 in detail by comparing it with phosphorylation obtained with two well-known opioids, the DAMGO peptide and buprenorphine. Partial versus full agonists and biased agonism which distinguishes two important signaling mechanisms, G-protein activation and beta-arrestin recruitment, are in the focus of receptorology. Based on these, the submitted communication is professionally important and timely. I found the article interesting and high quality so I definitely support its publication.

In order to further increase the value of this communication, I would like to consider, answer and accept the following questions and suggestions.

  • The chemical structure of the title compound should be included.

We included the chemical structure of all 3 compounds as a new Figure 1.

  • Fig 1. upper panel: the height of the fourth column appears to be high for buprenorphine (right). Compare with the corresponding column height of DAMGO (left). There, the spot of the immunoblotting is darker than that of buprenorphine.

The quantification was calculated from three independent experiments. We included a more representative western blot for S375 and T370.

  • In the confocal microscopy images shown in the second figure (Fig. 2), the internalized membrane receptors show a compact precipitation image and appear almost as nuclei. Why is this marking?

A explanation is stated now on page 4, line 90-92. “Furthermore, SR-17018 and DAMGO promote a robust internalization, which results in a receptor accumulation in the perinuclear recycling compartment. In contrast, buprenorphine failed to stimulate any detectable MOP endocytosis.“

  • 1-4. Spots of one phosphothreonine (pT340) are well visible for buprenorphine (Fig. 1), but are not present in any of the other figures. What could be the reason for this?

See answer to the second comment above.

Reviewer 2 Report

The work is well and concisely presented, with high scientific soundness. I guess, the unique kinetic profile of phosphorylation/dephosphorylation of MOP receptors exerted by SR-17018 might provide useful insights for the development of new drugs. Since we are dealing with a journal named "Molecules", I'm just wordering why the authors didn't show any chemical structures of the opioid agonists they discuss. I think they have to, with a brief intro included. Conclusion in simply "too short". I'm sure the authors can discourse more outcomes and perspectives. For mu opiod receptor I would use the acronym MOPr instead of MOP. 

Author Response

Reviewer 2 comments

Comments and Suggestions for Authors

The work is well and concisely presented, with high scientific soundness. I guess, the unique kinetic profile of phosphorylation/dephosphorylation of MOP receptors exerted by SR-17018 might provide useful insights for the development of new drugs.

  • Since we are dealing with a journal named "Molecules", I'm just wordering why the authors didn't show any chemical structures of the opioid agonists they discuss.

See answer to Reviewer 1 comment 1.

  • I think they have to, with a brief intro included. Conclusion in simply "too short". I'm sure the authors can discourse more outcomes and perspectives.

We extended our discussion on page 7.

  • For mu opiod receptor I would use the acronym MOPr instead of MOP. 

We always using the acronym “MOP” which is officially stated in the IUPHAR database.

Reviewer 3 Report

This manuscript clearly demonstrated the MOP phosphorylation and dephosphorylation stimulated by SR17018, a biased MOP agonist, and the authors compared it to those by DAMGO or buprenorphine. SR17018 induced gradual MOP phosphorylation from 20 sec to 30 min after stimulation (slower than DAMGO and similar to buprenorphine) and it retained for 60 min even if the agonist was removed (similar to buprenorphine). On the other hand, rapid MOP dephosphorylation was observed when naloxone were included in to the washout solution (similar to DAMGO). Interestingly, SR17018 promote a robust internalization 30 min after stimulation (similar to DAMGO).                                                                                                                                                                                                                                                                                                                                                                                                                                                                                                                                                                                                                                                                                                                                                                                                                                                                                                                                                                                                                                                                                                                                                                                                                                                                                                                                                                                                                                                                                                                                                                                                                                                                                                                                                                                                                                                              

The results of this study are clear and it includes beneficial information to understand the pharmacological profiles of biased agonist with reduced side effects like SR17018, but I found several problems as following.

Specific comments

  1. Are there any possible relationship between the delayed phosphorylation kinetics and the G-protein biased signaling? Authors had better include some discussion if any.
  2. Authors should describe the differences in the phosphorylation/dephosphorylation kinetics between the phosphorylation sites (e.g., pS375, pT370, pT376, pT379).
  3. Can author quantify the levels of internalization by SR17018 so that they can quantitatively compare it to that by DAMGO?

Author Response

Reviewer 3 comments

Comments and Suggestions for Authors

This manuscript clearly demonstrated the MOP phosphorylation and dephosphorylation stimulated by SR17018, a biased MOP agonist, and the authors compared it to those by DAMGO or buprenorphine. SR17018 induced gradual MOP phosphorylation from 20 sec to 30 min after stimulation (slower than DAMGO and similar to buprenorphine) and it retained for 60 min even if the agonist was removed (similar to buprenorphine). On the other hand, rapid MOP dephosphorylation was observed when naloxone were included in to the washout solution (similar to DAMGO). Interestingly, SR17018 promote a robust internalization 30 min after stimulation (similar to DAMGO)

The results of this study are clear and it includes beneficial information to understand the pharmacological profiles of biased agonist with reduced side effects like SR17018, but I found several problems as following.

Specific comments

  • Are there any possible relationship between the delayed phosphorylation kinetics and the G-protein biased signaling? Authors had better include some discussion if any.

We discussed that more in detail on page 7: “In fact, the selective engagement of different GRKs to differently activated MOP receptors could be a major source for biased signaling as it is the driving force for recruitment of arrestin isoforms 1 and 2 to the receptor [9] [10]. Thus, different GRK-mediated phosphorylation patterns should be taken into account during the development of new MOP agonists with beneficial side effect profile.

For many years, the biased signaling concept has been reduced to analysis of G protein signaling versus ß-arrestin 2 recruitment and the resulting bias factor has been proposed as predictor of the therapeutic window. SR-17018 is one candidate compound that was developed based on the biased signaling hypothesis [3]. While the initial study reported an extremely high bias factor in different G protein assays over ß-arrestin 2 recruitment, later work showed showed no statistically significant bias towards or away from any G protein activation [4]. Nevertheless, the present study revealed a unique MOP phosphorylation and internalization profile for SR-17018 that does not support the initial report of an extremely high bias factor.”

  • Authors should describe the differences in the phosphorylation/dephosphorylation kinetics between the phosphorylation sites (e.g., pS375, pT370, pT376, pT379).

The differences between the phosphorylation/dephosphorylation patterns are descript in detail on page 4 (line 86-88) and 5 (line 95-99).

It is stated on page 4: “S375 is the initial site of a hierarchical phosphorylation cascade. The following phosphorylation at T370, T379 and T376 requires priming S375 phosphorylation.”

It is stated on page 5: The DAMGO-induced phosphorylation was quickly reversed within 5 to 10 min after agonist removal. T370 and T379 were dephosphorylated immediately, whereas S375 and T376 dephosphorylation required between 10 to 20 min. In contrast, SR-17018-mediated MOP phosphorylation was retained for hours under otherwise identical conditions (Figure 4B).

  • Can author quantify the levels of internalization by SR17018 so that they can quantitatively compare it to that by DAMGO?

Immunocytochemistry is not the ideal method to quantify receptor internalization. Here new ELISA experiments would be necessary. Nevertheless, we could clearly show that SR-17018 is able to induces are robust internalization comparable like DAMGO.